# Effects of Two Physical Training Programs on the Cognitive Status of a Group of Older Adults in Chile

**DOI:** 10.3390/ijerph18084186

**Published:** 2021-04-15

**Authors:** Humberto Castillo Quezada, Cristian Martínez-Salazar, Sergio Fuentealba-Urra, Claudio Hernández-Mosqueira, Nelson Araneda Garcés, Fernando Rodríguez-Rodríguez, Yeny Concha-Cisternas, Edgardo Molina-Sotomayor

**Affiliations:** 1Facultad de Educación y Ciencias Sociales, Carrera de Educación Física, Universidad Andres Bello, Concepción 4030000, Chile; humberto.castillo@unab.cl (H.C.Q.); sergio.fuentealba@unab.cl (S.F.-U.); 2Departamento de Educación Física, Deportes y Recreación, Pedagogía en Educación Física, Facultad de Educación y Ciencias Sociales y Humanidades, Universidad de La Frontera, Temuco 4780000, Chile; claudiomarcelo.hernandez@ufrontera.cl; 3Departamento de Educación, Facultad de Educación y Ciencias Sociales y Humanidades, Universidad de La Frontera, Temuco 4780000, Chile; nelson.araneda@ufrontera.cl; 4IRyS Research Group, School of Physical Education, Pontificia Universidad Católica de Valparaíso, Valparaíso 2374631, Chile; fernando.rodriguez@pucv.cl; 5Escuela de Kinesiología, Facultad de Salud, Universidad Santo Tomás, Santiago 7750000, Chile; yenyf.concha@gmail.com; 6Pedagogía en Educación Física, Facultad de Educación, Universidad Autónoma de Chile, Santiago 7750000, Chile; 7Departamento de Educación Física, Pedagogía en Educación Física, Universidad Metropolitana de Ciencias de la Educación, Santiago 7750000, Chile; edgardo.molina@umce.cl

**Keywords:** older adult, aerobic capacity, muscular strength, cognitive state

## Abstract

Introduction: The effect of two physical training methods on older adults should be investigated in greater depth and its results shared with the community. Objective: To determine the effects of two types of physical training on the functional features associated with the cognitive state and the effect on a physiological mediator of growth hormone (IGF-1) in older women. Material and Methods: Quasi-experimental study that included 12 weeks of training in two groups divided into resistance and aerobic training. The study included a population of 113 women aged 69.39 ± 6.48 years from Talcahuano, Chile. All participants were randomly assigned to either group. The MINIMENTAL test was used to examine the executive functions of cognitive state and blood concentration of IGF-1, which was also used to examine neurotrophic factors. For the assessment of physical condition, an indirect test was used for the maximum mass displaced in one repetition (1RM) by the limbs and the TM6 test to estimate maximal oxygen consumption. Results: Significant differences between the groups with respect to the total score obtained in the MINIMENTAL test (EG1 = 28.13 ± 2.26; EG2 = 28.57 ± 1.83 and CG = 23.47 ± 2.80; ANOVA; *p* = 0.000) were observed. A post hoc analysis revealed no significant differences when examining executive functions individually between groups (Bonferroni; *p* > 0.05). An increase in the neurotrophic factor IGF-1 was also recorded in the training groups (EG1 *p* = 0.014 and EG2 *p* = 0.005). The pre- and post-test showed large differences in magnitude in the resistance training group (ES = 0.9; 20.41% change). Conclusion: Both workouts produce an overall improvement in the functions associated with cognitive status and increase blood concentrations of IGF-1 in older adults.

## 1. Introduction

The biological aging process of the human being is a phenomenon associated with functional limitations in the older adult [1], which has a global impact and is characterized by an increase in life expectancy. It is predicted that by 2050, a fifth of the world’s population will be over sixty, which will mean an increase in chronic and degenerative diseases, leading to the fragility and dependency of the people who suffer from them [2]. In addition, the aging of the population goes hand in hand with the decline in quality of life. According to Vives et al. [3], cognitive disorders and decline are some of the main causes of disability, incapacity, and reduced quality of life in older adults [4]. Among the cognitive disorders, senile dementia is the one that produces the greatest incapacity and dependency. Chile is not far removed from these issues, since the aging of its population is a fact verified as in other countries [4,5], indicating a deterioration in the quality of life [6]. The scientific evidence reveals that it is possible to prevent and/or delay the onset or effects of these diseases with suitable preventive treatments, including controlled physical activity or exercise, with the advantage of it being a low-cost solution to improve these people’s quality of life [3].

According to data from the World Bank [7], the average life expectancy in Chile is 79.57 years, which puts the country in second place in Latin America according to the National Institute of Statistics [8].

The World Health Organization [9] raises this age group’s need for attention and focuses on senile dementia, a situation covered in Chile as state policy [10,11]. The numbers are alarming, around 7% of adults over 65 years in Chile present some type of dementia [12].

Exercise attenuates cognitive decline and reduces dementia risk [13], achieving an increase in gray matter in the hippocampus in older adults that were in better physical condition [13]. Erickson et al. [14] suggest that the neurotrophic factors of the brain improve with physical activity, as this has a neuroprotective role.

Ding et al. [15] revealed that exercise increases insulin-like growth factor 1 (IGF-1) and is a mediator and inducer of brain-derived neurotrophic factor (BDNF), which improves the acquisition of learning and increases memory, i.e., it is a stimulant for cognitive capacity, which is exactly what is lost in old age. The role of IGF-I during exercise is associated with the action of BDNF, which is a critical modulator of synaptic plasticity in the brain, a mediator that improves an adult’s cognitive capacity. Both BDNF and IGF-1 increase the size of the hippocampus through exercise, positive effects on cognition [15], such as the prevention of neurodegenerative diseases, thus becoming a powerful way to improve brain health [14,15,16]. Therefore, it has also been shown that in older adults suffering from depression, aerobic and strength training have a positive effect on the psychological health and social behavior of patients [17]. Additionally, it improves memory and learning not only in young people but also advanced ages [18]. The aim of the present study was to determine the effects of two types of physical training on the cognitive state and a physiological mediator of IGF-1 in older women.

## 2. Materials and Methods

The study was conducted on a population of 113 older women in the commune of Talcahuano, Chile, with a sample of 63 older women with an average age of 69.89 years, divided into three groups in similar physical condition, with 20 people in each of the experimental groups and 23 individuals in the control group (Figure 1). A quasi-experimental study was designed for pre- and post-test. Non-probability sampling was used, with an allocation of random volunteers. Accounting for a desertion rate of 30.2%, the final sample of 44 older women was distributed equitably in three groups: the control group (CG), the aerobic training group (AG), and the resistance training group (RG). The following assessment protocols for maximum resistance were used: the indirect method described in Hardman’s protocol [19], aerobic capacity with the “Six-Minute Walk Test for Women” [20]. The cognitive state was measured with the “Mini-Mental State Examination-Folstein” (MMSE), validated in Chile by Quiroga [21], and the baseline of IGF1 was measured in the three groups at the beginning and end of the experiment. The team of researchers carried out the initial and final physical and cognitive evaluations, the blood samples to evaluate IGF1 were taken by a medical team. The present study was ascribed to the 2010 consort recommendation and approved by the ethics committee of the Universidad de Santiago de Chile (Ethics report N°140, 2017).

### 2.1. Intervention and Procedures

The intervention lasted 12 weeks, with a frequency of three times a week on alternate days (Monday, Wednesday, and Friday) in 60-min sessions, both groups carried out two adaptation weeks prior to the intervention, differentiating the intervention according to the groups Strength exercise program (RG). The sessions of this group had a session duration of (60) min, prior to the start of the program, two weeks of muscular adaptation were carried out with the aim of explaining the exercises that were developed on machines, the strength for different muscle groups, determine the maximum load according to Hardman [19], plan the training program, and learn the terminology according to the activities (intensity, repetitions, series, recovery, pause time). The intensity of the effort was determined according to the percentage of maximum load, and the exercises were divided into four exercises per session. (Table 1 and Table 2)

Aerobic Exercise Program (AG), prior to the start of the program, two adaptation weeks were carried out in order to know the Borg scale and learn to evaluate their heart rate. The sessions of this group had a duration of the central part of the session of (30–45) min, the activities consisted of entertaining dancing and walking exercises, they began with a low-intensity progression, then moderate and ended from moderate to vigorous, according to García [22]. During the activities, the participants maintain an intensity of effort between 50–70%, according to what was proposed by Karvonen and Vuorimaa [23] and considering between a 10–15 Borg’s subjective perception of effort scale [24], both parameters were supervised by the research team.

Strength exercise program (GF) The sessions of this group had a session duration of (60) min [9], prior to the start of the program, two weeks of muscular adaptation were carried out in order to explain the exercises that were developed on strength machines for different muscle groups, determine the maximum load according to Hardman [19], plan the training program and learn the terminology according to the activities (intensity, repetitions, series, recovery, pause time). The intensity of the effort was determined according to the percentage of maximum load, the exercises were performed according to what was proposed by Grande [25] and were divided into four exercises per session:

On day 1 of the week: Strength exercises against external resistance of upper limbs (MMSS); 3 lower limb exercises (MMII); 1 MMSS exercise, and 1 mid-body exercise.

On day 2 of the week: lower limb exercise (MMII); 2 MMSS counter-resistance strength exercises; 1 of the MMII, and 1 exercise of the middle zone of the body.

On day 3 of the week: exercise the middle zone of the body; 3 MMII strength versus resistance exercises and 1 core training mid-body exercise.

### 2.2. Data Analysis

The data analysis plan consisted of determining normality using the Shapiro-Wilk test and homoscedasticity (homogeneity of the variances) using Levene’s test. The variables of the samples are described using the mean and the standard deviation for continuous variables and with a confidence interval for the mean of 95% CI. A one-way ANOVA was applied for independent samples to determine the differences between the groups. This analysis was then complemented with a post hoc analysis with the Bonferroni test. In the intragroup comparisons (pre-post), the Student’s t or Wilcoxon tests were used to determine the magnitude of the intragroup differences. The value of the effect size was calculated using Cohen. To quantify the difference in the intergroup comparisons, Eta2 (η2) was applied. The statistical analyses were performed with statistical package SPSS version 23.0. The significance level was *p* < 0.05.

## 3. Results

Table 3 shows that the groups presented similar variances for all the study variables (UL, LL, VO_2_max, MMSE, IGF-1). The ANOVA test results made it possible to establish that the variables examined did not present significant pre-test differences.

In Table 4, statistically significant differences between the groups (CG-AG-RG) compared to the post-intervention muscular strength (MS) values can be seen. In the post hoc analysis, there were significant differences between the groups. The analysis of post-intervention means of the MS of the lower extremities revealed statistically significant differences, while the post hoc test established significant differences in two contrasts (CG-AG and CG-RG). Regarding the cognitive state (MMSE), significant differences were found between the groups regarding the MMSE scores. In the post hoc analysis, there were significant differences in two contrasts (CG-AG and CG-RG). The post-test results for the IGF-1 concentration showed the test yielded significant differences between the groups, and the post hoc analysis, showed significant differences were found between the CG-AG and the CG-RG.

In Table 5, significant differences appear in the pre- and post-test for the experimental groups AG and RG. With respect to the maximum resistance of muscular strength (MS), the RG experienced a significant change in the upper limbs (UL) and lower limbs (LL). Like the RG, the AG significantly increased the maximal resistance of muscular strength of the UL and in LL. When considering the results, aerobic capacity, expressed as the estimated maximal oxygen consumption (mL·kg^−1^·min^−1^) revealed significant differences only in the AG. The cognitive state, expressed in the MMSE score, presented significant changes for the AG and RG. In terms of IGF-1, there were differences in the pre- and post-test (*p* ≤ 0.05) for both the AG and the RG.

## 4. Discussion

The results indicate an improvement in the cognitive state and the IGF-1 concentration. Differences (*p* ≤ 0.05) were reported between the groups, with a large magnitude of effect for both the AG (effect size (ES) = 0.80) and the RG (ES = 1.0), consistent with Stein et al. [26] on the effects of exercise, resulting in the increase of IGF-1 levels and improvement in the cognitive state. Wall et al. [27] reported significant improvements in the executive functions (*p* = 0.02) with three months of intervention with aerobic exercises, also recording changes in the biomarkers (cortisol and IGF-1), finding a significant association with cognition. In terms of VO_2_max, the AG presented a greater change than the other two groups, endorsing the WHO recommendation [9] that aerobic exercise is the main method to maximize cognitive health in older adults. This was confirmed by the finding of Bouaziz [28], who observed that the groups that did aerobic activity presented higher levels of VO_2_max than the groups that did resistance or another type of training. The results obtained on the cognitive state are similar to the study by López et al. [29]. In the intervention, significant differences were found in favor of the intervened groups (AG, RG), considering the results by dimensions on the MMSE, mainly in the increase of processing capacity, retention, and evocation of verbal and visual information. Russo et al. [30] suggest the physical exercise improves cognitive functioning, especially of the executive functions and memory, and delays the onset of dementia. Erickson et al. [14] propose that aerobic exercise likely increases the size of the hippocampus in women of 70 and 80 years of age with slight cognitive decline because these people improved their spatial and verbal memory. The differences observed on the MMSE are consistent with Huang et al. [31], who concluded that training at an intensity of 35–50% significantly improved the VO_2_max, and at an intensity of 66–73%, the maximums of improvement in VO_2_max were obtained. Freudenderger et al. [32] concluded that at a greater VO_2_max, older adults will have better general cognitive functions such as memory and executive functions. Physical activity, but especially aerobic and resistance training, has a key role in the protection against cognitive decline and dementia through neuroplasticity processes [33].

In relation to the resistance training program for muscular strength, the results showed that the groups who took part in exercise significantly improved their post-intervention strength levels. Zoeller [34] states that there are still few studies that suggest that in the long-term, resistance training can bring cognitive benefits in older adults. The results showed that the AG experienced an average improvement of 53.1% with respect to the displaced mass (kg) in the estimation of a maximum repetition for the upper limb exercise (bench press) and 36.3% for the lower limb exercise (squat). The RG increased 66.8% in the estimation of a maximum repetition for the UL exercise and 65.4% maximum displaced mass estimated for the LL. Among the factors that can explain the increase in strength observed, for both the UL and LL in the RG, is that all the adults who took part had never undergone a training program, according to Candow et al. [35], who suggest that the subjects who have not trained strength for years respond favorably to resistance training. When analyzing the results of the muscular strength program and the effect they had on the cognitive state, significant differences were found in the three groups in cognitive state and IGF-1 (*p* < 0.001). These results are similar to those found by De Camargo et al. [36], who evaluated cognitive assessment with the Montreal Cognitive Assessment (MoCA) and reported a gain of 19% with (*p* = 0.01). Yoon et al. [37] noted significant increases in the levels of cognitive function and muscular strength after the application of resistance training; they used elastic bands on 30 older women with slight cognitive decline during a 12-week intervention period. The evidence presented makes it possible to assert that neuromuscular deterioration can be interrupted by training [38]. 

Gerlinger et al. [39] maintained that cognitive decline and the loss of muscle mass are characteristic of human aging. This can be avoided or mitigated by endogenous biological mechanisms such as the generation of IGF-1, which is induced by resistance training. In addition, Franco Martin et al. [40] suggest that “physical activity is being recognized as a highly protective factor, and it has been established today [41] as a promising psychosocial strategy for the protection of cognitive faculties”.

### 4.1. Strengths

This study contributes to the knowledge of professionals who work with the elderly population since it makes visible the effects of two strength and aerobic training programs and their effect on the cognitive state and plasma level of IGF1 in older women in Chile, which will allow diversifying the types of training in this population at risk, including resistance training.

### 4.2. Limitations

The present study was not without limitations, some variables were not considered, such as type of diet, nutritional status, and the socioeconomic level that could influence the results. With regard to the determination of VO_2_max, it was indirectly calculated from the test of 6 min of walking according to the Senior Fitness Test protocol and not directly by spiroergometry.

## 5. Conclusions

The program presented significant improvements in aerobic capacity and muscular strength, these types of training being a feasible and effective strategy to improve the functional and executive capacities of the cognitive state in older adults, being the GF against resistance the one that achieves the greatest changes in the mentioned variables. However, more studies are needed with interventions in older adults to deepen the potential of cardiorespiratory fitness and strength training in the variables of cognitive impairment and physiological indicators such as IGF1. Physical exercise should be considered as a valid therapeutic option to prevent and stop the functional and cognitive decline of older adults.

## Figures and Tables

**Figure 1 ijerph-18-04186-f001:**
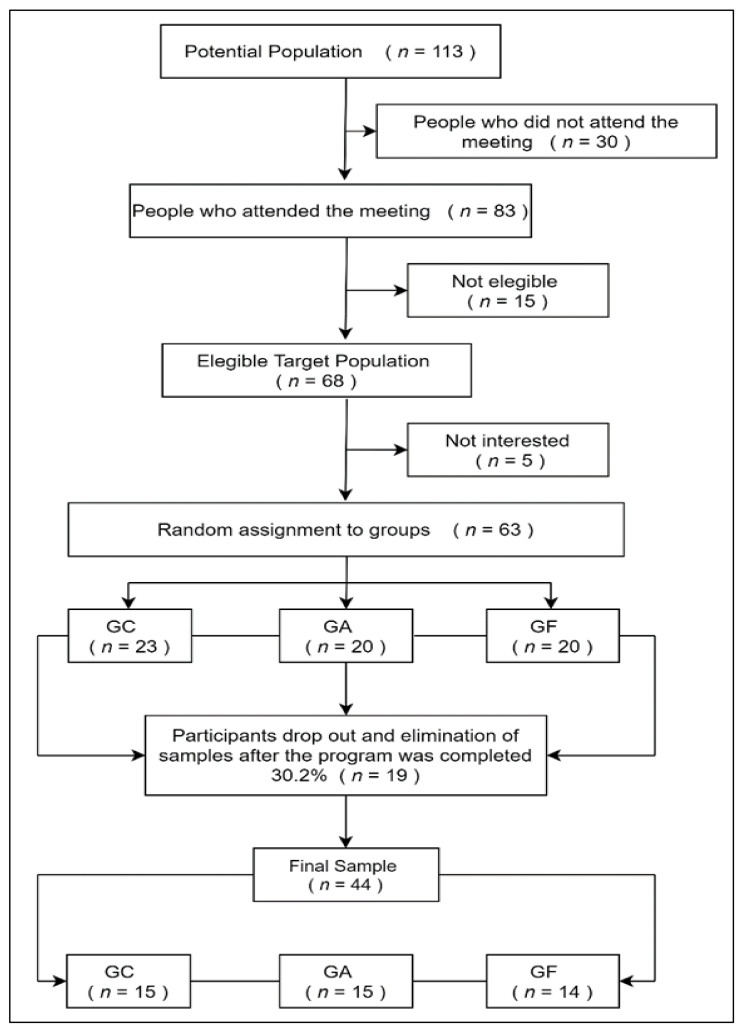
Sample recruitment scheme. Control Group GC: the aerobic training group GA: the resistance training group GF

**Table 1 ijerph-18-04186-t001:** Aerobic Exercise Program.

Weeks	1–2–3	4–5–6	7–8–9	10–11–12
Intensity	Under	Moderate	Moderate strong	Moderate to vigorous
Volume (min)	30	35	40	45
Parameters for Intensity control				
% Maximum Heart Rate	50	60	65	70
Subjective perception of effort (Borg scale)	10–11	12–13	14–15	14–15

**Table 2 ijerph-18-04186-t002:** Muscle Strength Exercise Program.

Weeks	1–3	4–6	7–9	10–12
Intensity % maximum load	50%	50–60%	60%	60%
Repetitions	10	10–12	12–15	15
N° Series	2	3	3–4	4
Recovery (Pause)	3 min	2.30–3 min.	2–2.5 min.	1.5–2 min.

**Table 3 ijerph-18-04186-t003:** Intergroup comparison of aerobic capacity (VO_2_max), maximum muscular strength (MS) of upper limbs (UL), lower limbs (LL), cognitive state (MMSE), and growth factor concentration (IGF-1). Pre-test.

Variables	CG (*n* = 15)		AG (*n* = 15)		RG (*n* = 14)		Comparison Intergroup ^§^		
	M ± SD	CI (95%)	M ± SD	CI (95%)	M ± SD	CI (95%)	F	*p* Value	Partial Eta^2^
MS UL (kg)	13.67 ± 1.91	12.61–14.73	14.40 ± 2.38	13.08–15.72	14.29 ± 2.58 Kg	12.79–15.78	0.437	0.649	0.1
MS LL (kg)	44.53 ± 3.50	42.59–46.47	47.20 ± 3.91	45.03–49.37	47.43 ± 3.88 Kg	45.19–49.67	2.701	0.079	0.4
VO_2_max. (mL kg^1^ min^−1)^	16.13 ± 1.80	15.14–17.13	16.02 ± 2.06	14.88–17.16	16.69 ± 1.10	16.06–17.32	0.625	0.540	0.2
MMSE (Score)	24.33 ± 3.52	22.38–26.28	26.80 ± 2.76	25.27–28.33	26.43 ± 2.65	24.90–27.96	2.912	0.066	0.5
IGF-1 (ng mL^−1)^	91.41 ± 23.10	78.61–23.10	107.88 ± 28.56	92.07–123.69	105.42 ± 39.08	82.86 -127.99	1.246	0.298	0.2

M ± SD = means and standard deviation; CI = confidence interval (CI 95%); ^§^ = one-factor multivariate analysis (Group); Eta^2^ = effect size.

**Table 4 ijerph-18-04186-t004:** Intergroup comparison of aerobic capacity (VO_2_max), maximum muscular strength (MS) of upper limbs (UL), lower limbs (LL), cognitive state (MMSE), and growth factor concentration (IGF-1). Post-test.

Variables	CG (*n* = 15)		AG (*n* = 15)		RG (*n* = 14)		Comparison Intergroup ^§^		
	M ± SD	CI (95%)	M ± SD	CI (95%)	M ± SD	CI (95%)	F	*p* Value	Eta^2^ Partial
MS UL (kg)	13.47 ±2.77 ^b.c^	11.93–15.00	21.40 ± 4.75 ^a.c^	18.77–24.03	23.57 ± 4.13 ^a.b^	21.19–25.95	26.443	0.000	0.7
MS LL (kg)	44.13 ± 4.69 ^b.c^	41.54–46.73	64.33 ± 18.46 ^a^	54.11–74.56	77.79 ± 26.07 ^a^	62.73–92.84	25.506	0.000	0.6
VO_2_max. (mL.kg^1^.min^−1)^	15.79 ± 2.15	11.60–16.98	16.59 ± 1.43	15.80–17.38	16.91 ± 1.02	16.32–17.50	1.885	0.165	0.2
MMSE (Score)	23.47 ± 2.80 ^b.c^	21.92–25.02	28.13 ± 2.26 ^a^	26.88–29.39	28.57 ± 1.83 ^a^	27.52–29.63	21.572	0.000	0.8
IGF-1 (ng.mL^−1)^	92.34 ± 22.80 ^b.c^	79.72–104.96	109.33 ± 28.74 ^a^	93.41–125.24	127.14 ± 51.56 ^a^	97.37–156.91	3.366	0.044	0.4

M ± SD = means and standard deviation; CI = confidence interval (CI 95%); ^§^ = one-factor multivariate analysis (Group); Eta^2^ = effect size; ^a,b,c^ = significant post hoc Bonferroni.

**Table 5 ijerph-18-04186-t005:** Intergroup comparison pre- and post-test of aerobic capacity (VO_2_max), maximum muscular strength (MS) of upper limbs (UL), lower limbs (LL), cognitive state (MMSE), and growth factor concentration (IGF-1).

Variables	CG (*n* = 15)			ES	AG (*n* = 15)			ES	RG (*n* = 14)			ES
	Pre	Post	*p*-Value		Pre	Post	*p*-Value		Pre	Post	*p*-Value	
VO_2_max. (mL·kg^1^·min^−1)^	16.13 ± 1.80	15.79 ± 2.15	0.451 ^b^	0.3	16.02 ± 2.06	16.59 ± 1.43	0.030 ^b^	0.7	16.69 ± 1.10	16.91 ± 1.02	0.105 ^a^	0.5
MS UL (kg)	13.67 ± 1.91	13.47 ± 2.77	0.745 ^a^	0.1	14.40 ± 2.38	21.40 ± 4.75	0.010 ^b^	1.2	14.29 ± 2.58	23.57 ± 4.13	0.011 ^b^	2.4
MS LL (kg)	44.53 ± 3.50	44.13 ± 4.69	0.531 ^a^	0.2	47.20 ± 3.91	64.33 ± 18.46	0.003 ^a^	0.9	47.43 ± 3.88	77.79 ± 26.07	0.001 ^b^	1.1
MMSE (Score)	24.33 ± 3.52	23.47 ± 2.80	0.316 ^a^	0.3	26.80 ± 2.76	28.13 ± 2.26	0.011 ^b^	0.8	26.43 ± 2.65	28.57 ± 1.83	0.004 ^b^	1.0
IGF-1 (ng·mL^−1)^	91.41 ± 23.10	92.34 ± 22.80	0.347 ^a^	0.3	107.88 ± 28.56	109.33 ± 28.74	0.014 ^a^	0.1	105.42 ± 39.08	127.14 ± 51.56	0.005 ^a^	0.9

M ± SD = means and standard deviation; CI = confidence interval (CI 95%); ^a^ = Student’s *t* test; ^b^ = Wilcoxon test; ES = effect size.

## Data Availability

The underlying research materials related to this paper are available from the corresponding author upon request.

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
