# Peer review of "Effects of Two Physical Training Programs on the Cognitive Status of a Group of Older Adults in Chile"

_ijerph, 2021, doi:10.3390/ijerph18084186_

Round 1

Reviewer 1 Report

The authors must describe the training plans carried out.

They must describe how the load control, volume, and intensity variations were carried out, as this is the key point for the results presented.

Author Response

We accept comments and modifications were included in the article

Reviewer 2 Report

Thank you for the opportunity to review this manuscript. This is a very fascinating study comparing three groups that examine training effects on cognition in older adults. The findings are important and novel and I enjoyed reading the paper. There are some areas that could be improved, and below are some comments to consider:

First line of abstract: You might give a quick background on the training methods – what types, and what do you mean by “greater depth”?

Lines 34-45 of abstract: what are EG1 and EG2 and CG? In terms of the type of exercise for the EG’s? Especially since you use AG and RG elsewhere in the manuscript

Line 63: Can you compare this prevalence to other similar/relevant countries for context?

The background is missing discussion of the types of training and their effect(s) on cognition. Please add that, perhaps before the aim sentence in line 75.

Methods: Just to clarify, your final sample was 44, after 63 were recruited from a population of 113? Can you provide any info on the commune – is it all women? Is 113 the total population or is that only the women? How does that compare to other areas?

Also, how exactly was the MMSE and IGF1 administered? In person? Self-report? Is there a reason you chose MMSE and 6MWT over other methods (e.g. MoCA, etc.)? More details are needed.

Line 115: Why is there a different font here?

Lines 119-129: Perhaps briefly list the differences between the groups, rather than just state there were differences

Line 151: You might add that the improvement occurred in the AG and RG groups to be specific since the CG decreased the MMSE.

Line 158: There were larger effects sizes for RG compared to AG, so it is interesting that you state aerobic exercise is the main method to maximize cognition. From your results, it appears that resistance training would maximize that more. VO2max would logically increase more in aerobic training compared to resistance. You might consider revision this section to reflect those points. Lines 174-176 seem more in line with your findings.

Line 170: This sentence mentions agreement with MMSE findings, then moves to VO2max findings. It might make more sense to break those into two sentences: one comparing MMSE and one comparing VO2max.

I would recommend adding consideration of strengths and weaknesses, perhaps after line 203. I enjoyed reading this study and think it would be beneficial to readers and the field to assess strengths/weaknesses as those could be considered in future research.

Author Response

(The authors gave the same response as above.)

Reviewer 3 Report

This study is experimental and divides people into two groups and therefore should have been registered under AllTrials and should also have been reported according to CONSORT principles.

Author Response

(The authors gave the same response as above.)

Round 2

Reviewer 3 Report

This has been extensively reviewed and I thank you for taking my points into consideration.